# OpenReview forum: "Concept Heterogeneity-aware Representation Steering"
_ICML.cc/2026/Conference — ICML 2026 regular_

### Official Review · Reviewer_a1Hk · 2026-03-06

**Soundness:** 3
**Presentation:** 3
**Significance:** 3
**Originality:** 3
**Overall Recommendation:** 4
**Confidence:** 4

**Summary:**

The paper proposes Concept Heterogeneity-aware Representation Steering (CHaRS), which reframes activation steering as an optimal transport (OT) alignment problem and argues that standard difference-in-means steering corresponds to a restrictive OT case. By modeling source/target activations as Gaussian mixture models, it solves a cluster-level OT problem and derives an input-dependent steering map via barycentric projection, yielding smooth, context-sensitive steering directions. Extensive experiments across safety/control tasks and multiple models show CHaRS improves behavioral control over global steering baselines

**Compliance With Llm Reviewing Policy:**

Affirmed.

**Final Justification:**

My concerns have been addressed. So I raise my score to 4.

**Key Questions For Authors:**

Please refer to the weakness.

**Limitations:**

Yes

**Strengths And Weaknesses:**

**Strengths**:

1.	Modeling activation steering as the optimal transport is an intellectually compelling perspective, and the paper provides a thorough and well-articulated theoretical treatment.

2.	The empirical evaluation spans a wide range of models, datasets, and tasks.

**Weaknesses**:

1.	The proposed method appears to partition the data into multiple clusters and to compute a query-adaptive steering vector based on these clusters. However, related work on adaptive representation steering (e.g., ACT [1]) has already explored a similar principle, which clusters steering directions constructed from positive/negative examples, whereas the present work clusters the positive/negative activation data directly. The authors are encouraged to elaborate more explicitly the conceptual and practical advantages of this design choice relative to ACT.

2.	Query-adaptive representation steering has been investigated across multiple domains [1–3]. Although the paper evaluates a broad set of datasets and models, it lacks substantive comparisons against other adaptive steering techniques. To strengthen the empirical claims, the authors should include additional, carefully controlled baselines and provide a persuasive comparative analysis against representative adaptive methods.

3.	Inference-time efficiency is a longstanding concern for representation steering methods. Because the proposed approach requires additional computation to form query-conditioned steering vectors at inference, the paper would benefit from a detailed analysis of the incremental inference cost.

[1] Wang T, Jiao X, Zhu Y, et al. Adaptive activation steering: A tuning-free llm truthfulness improvement method for diverse hallucinations categories. WWW 25.

[2] Wang W, YANG J, Peng W. Semantics-Adaptive Activation Intervention for LLMs via Dynamic Steering Vectors. ICLR 25.

[3] Parekh J, KHAYATAN P, Shukor M, et al. Learning to Steer: Input-dependent Steering for Multimodal LLMs. NIPS 25.

---

> ### Author Rebuttal · Authors · 2026-03-31
>
> Thank you for your thoughtful review and constructive feedback. Below we address your concerns.
>
> - W1: The reviewer is right that both ACT and CHaRS leverage multiple clusters for adaptive steering. However, the settings of ACT and CHaRS differ and our framework offers several concrete advantages over ACT's:
>
> | Aspect | ACT | CHaRS |
> |--------|-----|-------|
> | **Data requirement** | Limited to paired data: computes per-example $d = \bar{a}\_{\text{truthful}} - \bar{a}\_{\text{untruthful}}$ before clustering | Allows unpaired: clusters source/target activations independently, matches via OT (e.g., unrelated ADVBENCH vs. ALPACA) |
> | **Adaptation** | Independent direction $v^h_{c,l}$ and score $\alpha(1 - p_\theta(a) + \beta)$ per cluster, ignoring cross-cluster relations | Direction itself varies: $\hat{v}(x) = \sum_{i,j} w_{ij}(x) \cdot v_{ij}$, $w_{ij}(x) \propto P^*_{ij} \cdot k(x, a_i)$, yielding an interpretable, position-dependent vector field |
> | **Theory** | Empirically motivated | OT-grounded: DiM = OT map for equal-covariance Gaussians (Cor. C.3), generalized to GMMs via barycentric projection (Eq. 9); practical implementation simplifies full GMM-OT, but framework explains why global steering fails and paves way for future extensions/development (anisotropic mixtures, tighter transport approximations) |
>
> We will add a discussion in our revision on this comparison.
>
> - W2: We agree that comparison with adaptive steering methods is valuable. To this end, we directly compare against the Learning-to-Steer (L2S) framework [3] by integrating our method into it (CHaRS-L2S), where CHaRS provides structured, cluster-aware steering targets to train the network while keeping the L2S training and inference pipeline unchanged.
>
>
> **Table.** Safety steering evaluation of CHaRS-L2S vs. L2S on Qwen2-VL (MMSafetyBench). Lower is better for Unsafe-score, higher is better for ED-score. CHaRS-L2S uses CHaRS-derived structured steering targets within the L2S framework.
>
>
>
> | Metric | L2S | CHaRS-L2S | Improvement (%) |
> | :--- | :--- | :--- | :--- |
> | $\mathbb{E}_{p \geq 0.5}[\text{Unsafe-score}]$ $(\downarrow)$ | 0.0161 | **0.0150** | 6.83% |
> | $\mathbb{E}_{p \geq 0.7}[\text{Unsafe-score}]$ $(\downarrow)$ | 0.0085 | **0.0078** | 8.24% |
> | $\mathbb{E}_{p \geq 0.9}[\text{Unsafe-score}]$ $(\downarrow)$ | 0.0059 | **0.0053** | 10.17% |
> | ED-score $(\uparrow)$ | 0.566 | **0.605** | 6.89% |
>
>
> As shown in the table above, CHaRS-L2S consistently improves over L2S across all safety metrics on MMSafetyBench. The substantial improvement of CHaRS-L2S over L2S suggests that explicitly modeling concept heterogeneity provides a stronger signal for adaptive steering.
>
>
>
> - W3: Regarding inference efficiency, we evaluate the runtime overhead of CHaRS compared to standard steering (see the table below). We passed 100 harmless prompts and 104 harmful prompts respectively to Qwen2.5-7B-Instruct through VLLM.
>
>
> **Table.** Throughput Analysis of passing 100 harmless prompts through Qwen2.5-7B-Instruct.
>
> | Configuration | Total time (s) | Input Tokens |Input Throughput (tok/s)| Output Tokens | Output Throughput (tok/s)| Total Tokens | Throughput (tok/s) |
> |:--------------|:----|:--------|:-----|:-----|:-----|:-----|:-----|
> | No Steering | 5.60 | 4070 | 727.3 | 31796 | 5682.0 | 35866 | 6409.3 |
> | ActAdd | 5.31 | 4070 | 765.9 | 25120 | 4726.9 | 29190 | 5492.7 |
> | CHaRS-ActAdd | 5.34 | 4070 | 761.8 | 25243 | 4724.8 | 29313 | 5486.6
>
> **Table.** Throughput Analysis of passing 104 harmful prompts through Qwen2.5-7B-Instruct.
>
> | Configuration | Total time (s) | Input Tokens |Input Throughput (tok/s) | Output Tokens | Output Throughput (tok/s) | Total Tokens | Throughput (tok/s) |
> |:--------------|:----|:--------|:-----|:-----|:-----|:-----|:-----|
> | No Steering | 4.84 | 4307 | 889.0 | 17439 | 3599.6 | 21746 | 4488.6 |
> | ActAdd | 6.86 | 4307 | 627.6 | 46243 | 6738.4 | 50550 | 7365.9
> | CHaRS-ActAdd | 6.90 | 4307 | 623.9 | 45676 | 6616.1 | 49983 | 7239.9 |
>
> On Qwen2.5-7B-Instruct, CHaRS achieves comparable throughput to ActAdd, indicating minimal additional cost. This is because the extra computation, kernel weighting and cluster aggregation, is lightweight and does not require additional model forward passes. Overall, CHaRS maintains similar efficiency while providing improved steering performance.
>
> **Remark:** *We note that the throughput comparison between the base and jailbroken models is not entirely fair for harmful prompts, as refusal responses from the base model tend to be shorter, thereby reducing both the number of tokens generated and overall time required taking into account the asymmetry between the time for prefill and decoding.*
>
> We hope our responses have addressed your concerns. We would be happy to provide further clarification if needed.

---

> > ### Author Rebuttal · Reviewer_a1Hk · 2026-04-01
> >
> > I appreciate the authors' detailed rebuttal and the new experimental results, which have addressed most of my initial concerns. I am therefore increasing my score. I encourage the authors to thoroughly incorporate the additional comparisons and the more comprehensive discussion of related work into the final version to further strengthen the paper's impact.

---

> > > ### Author Response · Authors · 2026-04-01
> > >
> > > Thanks for your response, and we appreciate your endorsement. We will incorporate your suggestions, including the additional comparisons and a more comprehensive discussion of related work, into the final version of the paper.

---

### Official Review · Reviewer_pCdi · 2026-03-11

**Soundness:** 4
**Presentation:** 4
**Significance:** 3
**Originality:** 3
**Overall Recommendation:** 4
**Confidence:** 4

**Summary:**

In this work the authors present a concept-heterogeneity-aware representation steering method using Optimal Transport techniques. They begin with the observation that existing difference-in-means steering methods equate to the OT distance between Gaussians with identical covariance structure, and drop these assumptions by adopting a mixture model of gaussians with different (isotropic) covarianes. They clearly go walk the reader through the extension of OT to multimodal gaussians, introducing Gaussian mixture couplings, the subsequent Mixture Wasserstein distance, and then derive the barycentric mapping. Next they describe how the mixture coupling is estimated in practice with Entropy-Regularized OT for the cluster centroid matching, soft assignment using a RBF kernel, and closed-form matching for the Gaussians (although they require mean and isotropic covariance estimation). Finally, they highlight how their transport plans are upper bounded in rank by the number of clusters, allowing them to take principle components when steering. From an experiment side, they evaluate their method on jailbreaking, toxicity mitigation, and image generation steering.

**Compliance With Llm Reviewing Policy:**

Affirmed.

**Final Justification:**

The rebuttal answered my questions but highlighted some limitations of the paper: empirical results don't show the method is uniformly better (which is okay) and the motivation of a principled framework is somewhat lost in relaxations and approximations (isotropic gaussianity, approximate transport map). These relaxations are also reasonable but detract from the motivation of the formal OT treatment. Overall, I still think this paper is a weak accept for ICML.

**Key Questions For Authors:**

* What exactly are the ActAdd and DirAbl interventions? At first I thought they were baselines but it seems to be two different steering methodologies over which you apply CHaRS. It might be good to introduce them in the main body (or refer to them in the appendix)
 * On line 244, should the terms "transport aware effective prior" and "empirical cluster size prior" be swapped? (as in, you are replacing the transport aware effective prior with an empirical approximation)

**Limitations:**

yes

**Strengths And Weaknesses:**

## Strengths

 * The theoretical motivation is intuitive and justifiable and seems like the natural next step in steering research.
 * The explanation of the method and OT concepts used is clear and easy to understand.
 * The empirical results demonstrate the efficacy of the method.

## Weaknesses
 * If the motivation for this work is concept heterogeneity and the general complex nature of model representations, the isotropic gaussian assumption seems very restrictive. For instance, if these concepts and steering vectors lie on a manifold, I would expect the covariance structure to be low-rank rather than isotropic (isotropic might have some beneficial smoothing properties though). Can you justify this choice, and I understand it would be too hard to do experiments but are there similar complexity methods for estimating low-rank covariance structures?
 * Shouldn't the optimal transport map and barycentric projection already define an interpolation between clusters? If you go through all this work to ground your steering in the barycentric projection, why do you need to steer at higher strengths (alpha > 1)? (I understand in past work the community sweeps over strengths, but I feel like recent literature [1] has advocated for restraint in steering and I would assume doing a smarter approach like this would avoid the need for large steering magnitudes
 * While Figure 2 bottom is pretty convincing for the performance of CHaRS, on the top two figures, it appears as though CHaRS causes more severe degradation in ClipScore while Linear-ACT keeps the semantics about the same. It actually seems as though your method is steering "too well" and is destroying the semantics of the original sentence (which isn't the worst thing). Would you agree with this interpretation of the figure? If so, it may make sense to change the discussion of the result.

[1]: Fel, T., Wang, B., Lepori, M. A., Kowal, M., Lee, A., Balestriero, R., ... & Wattenberg, M. (2025). Into the rabbit hull: From task-relevant concepts in DINO to minkowski geometry.

---

> ### Author Rebuttal · Authors · 2026-03-31
>
> Thank you for your thoughtful review and constructive feedback. Below we address your concerns.
>
> - W1: Although we agree that the isotropic assumption is restrictive and that a low-rank covariance structure would be more natural, our choice is motivated by simplicity and practical stability. Estimating even low-rank covariances per cluster from limited samples in high dimensions is likely to yield low signal-to-noise ratio that can degrade steering quality or make tuning harder. That said, low-rank or diagonal-restricted alternatives do exist at comparable complexity. We view this as a natural and promising extension. To test this hypothesis, we experiment on a new settings, where $A$ is derived from diagonal covariance matrices. We observe that the isotropic Gaussian assumption yields higher ASR compared to diagonal covariance setting indicating possible low quality of covariance estimation as expected. However, CHaRS outperforms the baseline in both settings, indicating the importance of heterogeneity modeling in general. Exploring full covariance variants of CHaRS is a compelling future work under a more relaxed data constraints.
>
> Table: ASR for CHaRS with different covariance structures, tested on Qwen2.5-7B-Instruct. Best ASR is **bolded**.
> | Configuration | $K$ | ASR |
> |:--------------|:----|:-----|
> |Isotropic|1 |91.35
> |Isotropic |5|**95.19**
> |Diagonal Covariance| 1 | 85.58
> |Diagonal Covariance| 5 | **86.54**
> - W2: This is a thoughtful point. In principle, the barycentric projection defines the "correct" transport and $\alpha = 1$ should indeed suffice. In practice, however, the transport map is approximate as detailed in Section 3.2. Those relaxations collectively attenuate the effective steering magnitude, so tunable $\alpha$ compensates for this. We agree with the broader sentiment that principled steering should reduce reliance on large magnitudes, and indeed our results show that CHaRS achieves strong style induction at lower strengths than Linear-AcT (peak at $\lambda \in [0.8, 1]$ vs. much larger values), which we view as evidence that better modeling of heterogeneity moves in precisely this direction.
>
> - W3: We agree with the interpretation. CHaRS steers more effectively at lower strengths, so it can appear to cause more severe degradation. However, this merely reflects earlier and stronger overriding of the original semantics rather than purely excessive degradation. In Figure 2 top, CLIPScore for CHaRS begins to decrease shortly after 0-Shot Classification Score peaks, indicating that strong alignment with the target style is achieved just as semantic changes emerge. In contrast, for Linear-AcT, the 0-Shot Classification Score continues increasing even after CLIPScore has decreased substantially. The Pareto front (Figure 2 bottom) makes this trade-off explicit by demonstrating that at matched levels of style induction, CHaRS preserves semantics better, achieving a better overall trade-off. We will revise the discussion to acknowledge this nuance more explicitly: CHaRS is more potent per unit of steering strength, which is advantageous when strength is tuned appropriately but can be destructive if applied at the same magnitude as weaker methods.
> - Q1: We thank the reviewer for pointing this out. Although we describe ActAdd and DirAbl in Appendix D.1 and D.2, we did not explicitly refer to them in the main text. We will clarify their roles and how CHaRS integrates with them in the revision.
> - Q2: Reviewer's confusion is indeed valid. We realized the transition from $p_k$ being the theoretical quantity $\sum_{l}\gamma^*_{kl}$ to empirical cluster sizes in line 190 (second column) is not highlighted enough and we will improve the writing in our revision. Also, we say $\sum_{j}P_{kj}$ is transport-aware under the OT objective in Eqn. (10).
>
> We hope our responses have addressed your concerns. We would be happy to provide further clarification if needed.

---

> > ### Author Rebuttal · Reviewer_pCdi · 2026-04-01
> >
> > Thank you for the responses!

---

> > > ### Author Response · Authors · 2026-04-01
> > >
> > > Thank you for your engagement and for acknowledging that all concerns were fully resolved.
> > >
> > > To briefly summarize: we justified the isotropic assumption with new diagonal covariance experiments showing CHaRS consistently outperforms baselines regardless of covariance structure; we clarified that $\alpha \ne 1$ compensates for practical approximations.
> > >
> > > We thank the reviewer once again for your positive assessment. We hope that this clarification, together with the experimental evidence provided in the rebuttal, might be taken into account when considering whether an increased score would better reflect your updated evaluation of our paper.

---

### Official Review · Reviewer_3Qaq · 2026-03-11

**Soundness:** 3
**Presentation:** 3
**Significance:** 2
**Originality:** 3
**Overall Recommendation:** 4
**Confidence:** 3

**Summary:**

This paper proposes Concept Heterogeneity-aware Representation Steering (CHaRS), an inference-time activation steering method for LLMs (and diffusion models) that replaces a single global steering direction (e.g., difference-in-means) with an input-adaptive steering field. The key idea is that many “concepts” (e.g., harmfulness vs. harmlessness, toxicity vs. neutral, style vs. non-style) are heterogeneously represented in activation space, often forming clusters that correspond to latent sub-concepts or contexts, making global steering brittle.
The authors formalize classical difference-in-means steering as an optimal transport (OT) map between two Gaussians with equal covariance (yielding a pure translation). To relax the unimodal assumption, they model source/target activation distributions as Gaussian mixture models (GMMs) and formulate alignment as a discrete OT problem over clusters (estimated via k-means), solved with entropic OT (Sinkhorn). From the resulting transport plan, they derive a barycentric-projection-inspired transport map that produces a smooth, kernel-weighted combination of cluster-to-cluster shift vectors as the steering vector for each input token representation.
They further propose CHaRS-PCT, exploiting a low-rank structure of the set of cluster-level steering vectors to factorize and truncate the steering field using PCA / Principal Component Thresholding, aiming to reduce the number of effective directions while maintaining performance.
Experiments are reported on (i) LLM jailbreaking (improving Attack Success Rate while tracking tinyBenchmark utility), (ii) toxicity mitigation in a sequential steering setting compared to Linear-AcT (transporting activations), and (iii) diffusion image style control (e.g., cyberpunk, sketch) on FLUX.1, measured by CLIP-based style classification and content preservation.

**Compliance With Llm Reviewing Policy:**

Affirmed.

**Final Justification:**

I will keep my initial rating as 4: Weak accept

**Key Questions For Authors:**

1. Ablation: OT vs. non-OT cluster matching. If you replace Sinkhorn OT with simpler alternatives (e.g., nearest-centroid matching, Hungarian matching, or uniform coupling), how much performance drops across tasks?
2. Layer/token intervention choices. CHaRS defines an input-dependent map per representation. Which layers and token positions are intervened on for each task, and how sensitive are results to these choices (e.g., last token only vs. all tokens; single layer vs. multiple layers)?

**Limitations:**

No. The “Impact Statement” is not adequate. The paper should explicitly discuss risks of enabling harmful capability amplification (notably jailbreaking), sensitivity/tuning issues, potential distribution shift failure modes, and responsible release guidelines (e.g., restricting steering vectors/datasets or focusing on defensive steering).

**Strengths And Weaknesses:**

Strengths
1. The methodological motivation is technically grounded: the paper cleanly connects difference-in-means steering to a specific OT special case (Gaussian-to-Gaussian with equal covariance), then generalizes to mixture models and a component-level OT coupling. The derivation from coupling to a deterministic map via barycentric projection is reasonable and consistent with OT literature.
2. The proposed practical algorithm (k-means clusters → Sinkhorn coupling → RBF gating → weighted shift vectors) is straightforward and implementable; the inclusion of implementation details (e.g., Sinkhorn steps, how CHaRS plugs into ActAdd/DirAbl) supports credibility.
3. The paper is clearly organized, with a coherent narrative from (i) limitations of global steering to (ii) OT interpretation to (iii) GMM/cluster OT formulation and (iv) practical approximation leading to CHaRS, followed by experiments and ablations.

Weaknesses
1. In the final deployed method, the theoretical GMM-OT story is simplified to isotropic covariances and effectively centroid translation costs (squared Euclidean distance between centroids). This is a sensible engineering choice, but it weakens the claim that the method is truly performing GMM-OT beyond a “clustered translation field,” and it is unclear when the OT component adds value beyond simpler cluster matching heuristics.
2. For vision (style control), the evaluation is primarily CLIP-based classification and CLIPScore; more human evaluation or stronger perceptual metrics would better support the claim of superior style/content trade-offs.

---

> ### Author Rebuttal · Authors · 2026-03-31
>
> - W1: We appreciate this observation. The isotropic covariance assumption is a deliberate simplification within the GMM-OT hierarchy: exact GMM-OT admits no closed-form solution, so approximations are necessary in practice. Mixtures of isotropic Gaussians remain strictly richer than a single Gaussian. Moreover, the framework is modular: each component-wise translation can in principle be extended to the full affine map $T(x) = m_2 + A(x - m_1)$, which is optimal under Gaussian-to-Gaussian transport (cf. Eqn. (2)). However, covariance estimation is observed to be hard with very limited samples (especially after clustering). For proof of concept, nonetheless, we test CHaRS with $K=1$ versus $K=5$ employing estimations of diagonal covariances.
>
> Table 1: ASR for CHaRS with different covariance structures, tested on Qwen2.5-7B-Instruct. Best ASR is **bolded**.
> | Configuration | $K$ | ASR |
> |:-|:-|:-|
> |Isotropic|1 |91.35
> |Isotropic |5|95.19
> |Diagonal Covariance| 1 | 85.58
> |Diagonal Covariance| 5 | 86.54
>
> - W2: Following the reviewer’s suggestion, we conduct human evaluation to assess both style matching and content fidelity. The study includes four subsections: Cyberpunk style, Sketch style, and content matching under each style. Our data collection employs a two alternative forced choice (2AFC) test, where participants are shown paired images generated by CHaRS and Linear-AcT, and asked to choose the one that better matches the target style (style descriptions provided) or content (image caption provided).
> We report results as the preference rate of each method over the other. As the task is binary and symmetric, the percentages on both sides sum to 1, and both methods share the same variance.
>
> Table 2. Human evaluation of CHaRS vs. Linear-AcT.
> | Percentage (%) | Linear-AcT | CHaRS | Std Dev
> |:-|--|:--|:-|
> | Style (Cyberpunk) (&uarr;) | 45.68% | **54.32%** | 3.39
> | Style (Sketch) (&uarr;) | 21.06% | **78.94%** | 2.88
> | Content (Cyberpunk) (&uarr;) | 40.23% | **59.77%** | 3.17
> | Content (Sketch) (&uarr;) | 36.89% | **63.11%** | 4.54
>
> Table 3: The LPIPS score (lower is better) and its variance are presented for each method.
> | Style       | Method | Mean (&darr;) | Variance |
> |:------------|:---------|:----|:----|
> | Cyberpunk   |Linear-AcT|0.70   |0.0036
> |Cyberpunk    |CHaRS     |**0.67**   |0.0037
> |Sketch       |Linear-AcT|0.74   | 0.0038 |
> |Sketch       |CHaRS     |**0.69**   |0.0032
>
> - Q1: The entropy-regularized OT coupling further enforces marginal constraints that respect cluster mass, yielding geometrically principled correspondences that simpler alternatives (e.g., nearest-centroid or Hungarian matching) do not guarantee under imbalance. That said, for Question 1, to evaluate the effects of the coupling strategy for cluster matching, we conduct an ablation study comparing 3 different coupling methods: Nearest Neighbour, Uniform Coupling, and Optimal Transport (OT) Coupling. All experiments are conducted on Qwen2.5-7B-Instruct.
> We note that at $K=1$, coupling is a trivial problem leading to the same ASR of 91.35. Therefore, we focus our comparison on $K \neq 1$ where the coupling strategy plays a meaningful role. We also report the ASR for each coupling strategy at $K = 5$, which is the value of $K$ that yields the best ASR for OT Coupling.
>
> Table 4: Ablation study of the effects of different coupling strategies for cluster matching. Best ASRs are **bolded**. For reference, all methods achieve an identical ASR of 91.35 at $K=1$, where the coupling choice has no effect.
> | Coupling | ASR at $K=5$ | Best ASR ($K \neq 1$)| Best $K$ ($K \neq 1$)|
> |:---------|:------------|:----|:----|
> |Nearest Neighbour |76.92|94.23|7 |
> |Uniform Coupling|42.31|87.50|8|
> |OT Coupling|**95.19**| **95.19**| 5|
>
> - Q2: For token positions, similar to previous works [1, 2], we intervene on all tokens. For layer choices, we run experiments on Qwen2.5-7B-Instruct and report the ASR for both single-layer and multi-layer strategies in Table 5. To extend our results, we further test the effect on ASR if we consider the mean of activations from all token positions when constructing the candidate intervention vectors. We believe that as such averaging can dilute the cluster structure otherwise observed in the set of last tokens, CHaRS offers no significant benefits over the baseline with $K=1$.
>
> Table 5: Ablation study of the effects of different intervention choices on ASR. Best ASRs are **bolded**.
> | Intervention Layer | Token Position | $K$  | ASR |
> |:---------|:----|:----|:----|
> |Single Layer| Last Token |1 |91.35
> |Single Layer| Last Token |5 |**95.19**
> |Sequential  | Last Token |1| 79.81 |
> |Sequential  | Last Token |5| **82.69**
> |Single Layer| All Tokens |1 |**34.62**
> |Single Layer| All Tokens |9 |**34.62**
>
> **References:**
> [1] "Controlling Language and Diffusion Models by Transporting Activations". ICLR 2025.
> [2] "Refusal in Language Models Is Mediated by a Single Direction". NeurIPS 2024.

---

### Decision · Program_Chairs · 2026-04-30

**Decision:**

Accept (regular)

**Comment:**

In this paper, the authors propose CHaRS, a representation steering method that casts steering as a Gaussian mixture optimal-transport (OT) alignment problem. They base it on the observation that difference-in-means (DiM) steering is the OT map between equal covariance Gaussians and motivating a natural generalization of DiM over mixture components. Experiments include jailbreaking, sequential toxicity mitigation, and diffusion style control. All three reviewers settled on Weak Accept (4) after rebuttal, with two marking their concerns fully resolved. The rebuttal delivered the diagonal-covariance ablation on Qwen2.5-7B (ASR 86.54 vs isotropic 95.19), a CHaRS-L2S integration on MMSafetyBench (6.8-10.2\% improvement across safety metrics), an ACT-vs-CHaRS design comparison, and a throughput table showing under 1\% overhead over ActAdd. The main remaining weakness, noted by two reviewers, is that the deployed method collapses to isotropic-covariance centroid translations. For these reasons, I am leaning towards weak accept.

Potential Hallucinated Citation: the Turner et al. 2023 reference on p.11 (arXiv:2308.10248) lists author names and title that do not match the actual arXiv record for that ID.